# Stability of Estimated Premorbid Cognitive Ability over Time after Minor Stroke and Its Relationship with Post-Stroke Cognitive Ability

**DOI:** 10.3390/brainsci9050117

**Published:** 2019-05-22

**Authors:** Caroline A. McHutchison, Francesca M. Chappell, Stephen Makin, Kirsten Shuler, Joanna M. Wardlaw, Vera Cvoro

**Affiliations:** 1Centre for Clinical Brain Sciences, University of Edinburgh, Chancellors Building, Edinburgh Bioquarter, 49 Little France Crescent, Edinburgh EH16 4SB, UK; caroline.mchutchison@ed.ac.uk (C.A.M.); F.Chappell@ed.ac.uk (F.M.C.); Stephen.Makin@glasgow.ac.uk (S.M.); kshuler@pennstatehealth.psu.edu (K.S.); Vera.Cvoro@ed.ac.uk (V.C.); 2Centre for Cognitive Ageing and Cognitive Epidemiology, Department of Psychology, University of Edinburgh, 7 George Square, Edinburgh EH8 9JZ, UK; 3Institute for Cardiovascular and Medical Sciences, University of Glasgow, Glasgow G12 8TA, UK; 4Neuroscience Institute Penn State University, College of Medicine, Penn State Hershey, Hershey, PA 17033, USA; 5UK Dementia Institute at the University of Edinburgh, Edinburgh EH16 4SB, UK; 6NHS Fife Victoria Hospital, Kirkcaldy KY2 5AH, UK

**Keywords:** premorbid intelligence, NART, cognition, stroke

## Abstract

Considering premorbid or “peak” adult intelligence (IQ) is important when examining post-stroke cognition. The stability of estimated premorbid IQ and its relationship to current cognitive ability in stroke is unknown. We investigated changes in estimated premorbid IQ and current cognitive ability up to three years post-stroke. Minor stroke patients (NIHSS < 8) were assessed at one to three months, one and three years’ post-stroke. The National Adult Reading Test (NART) and Addenbrooke’s Cognitive Examination-Revised (ACE-R) were used to estimate premorbid IQ (NART IQ) and current cognitive ability respectively at each time-point. Baseline demographics, vascular and stroke characteristics were included. Of the 264 patients recruited (mean age 66), 158 (60%), 151 (57%), and 153 (58%) completed cognitive testing at each time-point respectively. NART IQ initially increased (mean difference (MD) = 1.32, 95% CI = 0.54 to 2.13, *p* < 0.001) before decreasing (MD = −4.269, 95% CI = −5.12 to −3.41, *p* < 0.001). ACE-R scores initially remained stable (MD = 0.29, 95% CI = −0.49 to 1.07, *p* > 0.05) before decreasing (MD = −1.05, 95% CI = −2.08 to −0.01, *p* < 0.05). Adjusting for baseline variables did not change the relationship between NART IQ and ACE-R with time. Increases in NART IQ were associated with more education. For ACE-R, older age was associated with declines, and higher NART IQ and more education was associated with increases. Across 3 years, we observed fluctuations in estimated premorbid IQ and minor changes in current cognitive ability. Future research should aim to identify variables associated with these changes. However, studies of post-stroke cognition should account for premorbid IQ.

## 1. Introduction

Cognition in childhood determines cognitive ability in later life in healthy persons [1] and influences cognition in those with dementia [2]. Therefore, it is important to consider peak adult intelligence (IQ), also referred to as “premorbid IQ”, when assessing cognitive ability in later life to allow for the examination of cognitive decline [3]. Premorbid IQ refers to an individual’s level of intellectual functioning prior to declines associated with ageing, neurological events, or both [4]. Declining cognitive ability in the years preceding the stroke can be measured with tests, such as the Informant Questionnaire on Cognitive Decline (IQCODE) [5], but post-stroke cognitive impairment is common and premorbid IQ is an important predictor [6,7]. Changes in cognitive ability are clinically important, therefore, the availability of stable and reliable estimates of premorbid IQ are vital so that we can accurately examine risk factors for post-stroke cognitive decline. 

As actual IQ scores from early adulthood are rarely available, methods of estimation have been developed. Widely accepted measures, such as the National Adult Reading Test (NART) [8], assess vocabulary and pronunciation skills, which are considered as “crystallized intelligence” and remain unaffected by neurological events or impairment [8]. Using the NART, premorbid IQ (NART IQ) can be estimated based on the number of correct responses (NART score). The NART has been shown to relate well to current IQ scores in healthy individuals [9] and to childhood IQ for individuals with and without dementia [2]. It has been used in different neurological disorders [10] and has been shown to have good consistency over periods up to 7.5 years [11,12]. However, some evidence suggests that it can over- or under-estimate [9] and is less reliable in some patient populations [13]. 

To our knowledge, the stability of the NART after stroke has not been examined. Furthermore, there is a scarcity of studies examining longitudinal changes in premorbid IQ (which should be stable) and current cognitive ability (which declines with age). 

We aimed to investigate:
Whether the NART is a valid test of premorbid IQ after stroke by examining the relationship between time after stroke with changes in NART IQ, and current cognitive ability (i.e., how do estimated premorbid IQ and current cognitive ability scores change between follow-up time-points). We will also examine whether the relationships with time remain after controlling for baseline demographic and stroke characteristics. Due to the obvious relationship between age and time, we examine the relationship between age, estimated premorbid IQ and cognitive ability at one to three months post-stroke. Whether certain baseline demographic or stroke characteristics predict changes in NART IQ and current cognition scores between two time-points post-stroke.


## 2. Materials and Methods

### 2.1. Recruitment

Patients (over 18 years) with recent minor ischemic stroke were recruited into the Mild Stroke Study II (MSSII) between May 2010 and May 2012. This prospective study consecutively recruited in- and outpatients with minor ischemic stroke admitted to the Lothian Stroke Services, Scotland. Patients who were unable to consent (e.g., lacked capacity due to cognitive impairment), medically unlikely to participate in long-term follow-up and had aphasia were excluded (details previously described) [7,14].

Minor ischemic stroke was defined as sudden onset of focal neurological symptoms lasting >24 hours, a National Institutes of Health Stroke Scale (NIHSS) score of <8, that was not expected to result in long-term dependency (modified Rankin score (mRS) ≤ 3). Based on the clinical stroke syndrome and MRI appearance of the acute lesion, the index stroke was classified as “cortical” or “lacunar” [15]. MRI including diffusion-weighted imaging (DWI) was performed using a 1.5 Tesla GE Sigma HDxt scanner [14]. White matter hyperintensities (WMH) were rated using the Fazekas score [16], blind to clinical and cognitive information [14].

### 2.2. Follow-Up Assessments

Following recruitment and baseline clinical assessment, patients were invited for cognitive assessment at one to three months, and one and three years’ post-index stroke. Only those who were seen for in-person assessment were offered cognitive assessment. Follow-up at one to three months and 1 year were performed in the hospital. At three years’, participants unable to attend the hospital were offered a home visit or telephone interview.

We assessed premorbid IQ using the NART and current cognitive ability using the Addenbrooke’s Cognitive Examination–Revised (ACE-R) [17] at all three time-points. The NART consists of 50 phonetically irregular words, which participants read aloud. The ACE-R is a multi-domain cognitive screening tool used to identify those with cognitive impairment. Assessments were completed by trained researchers, experienced in the administration of these neuropsychological screening tools. In addition, we collected information relating to socio-demographic, vascular risk factors, lifestyle variables, and recurrent vascular events including stroke [6].

This study was approved by the Lothian Ethics of Medical Research Committee (REC 09/81, 101/54) and the NHS Lothian Research and Development Office (2009/W/NEU/14). All participants gave written informed consent at each time-point.

### 2.3. Statistical Analysis

Several methods of calculating premorbid IQ using NART errors scores are available. We calculated “NART IQ” based on the number of correctly pronounced words on the NART using the equation: 127.7 − 0.826 × (50 − NART score) [8]. Mild and severe cognitive impairment were defined as ACE-R scores 83–88 and ≤82, respectively [17].

We used Wilcoxon rank-sum test (*W*), *t*-test and chi-square tests to compare those with NART IQ and ACE-R versus those without (questionnaire only and no follow-up), tested at each time-point based on demographics, lifestyle, and stroke characteristics.

#### 2.3.1. Aim 1

We used a series of paired t-tests to examine whether NART IQ and ACE-R scores significantly differed between one to three months and one year, and one and three years (mean differences (MD)). We used linear mixed models to examine whether the associations between time and NART IQ or ACE-R scores were affected by baseline characteristics (i.e., when adjusting for baseline factors, does the association between time and NART IQ or ACE-R scores differ from that seen in paired *t*-tests). We included baseline demographic (age, sex, and years of education), stroke characteristics (stroke subtype (cortical/lacunar), stroke severity (NIHSS score, 0–8), Fazekas score (0–6)), and vascular and lifestyle risk factors (hypertension (yes/no), smoking status (current and less than 1 year ex-smoker/more than 1 year ex-smoker and never smoked)) factors. However, our models showed poor model fit: Model residuals were non-normal (Appendix A) and could not be improved through standard data transformations. Therefore, results should be interpreted with caution. Due to the obvious relationship between age and time, we explored the relationships between NART IQ and ACE-R with age using linear regression.

#### 2.3.2. Aim 2

We examined predictors of change in NART IQ and ACE-R scores between each time-point by including an interaction with time in our linear mixed effect models. To increase power, we examined the interaction between time and each baseline demographic and stroke variable in separate models resulting in a series of strongly related and overlapping models. It is important to note that we hoped to identify effect which may be of interest in future research, therefore, we did not adjust for multiple comparisons [18].

## 3. Results

### 3.1. Patient Characteristics

At baseline, 264 patients with minor ischemic stroke were included (mean age 66.40, SD 11.84, 154 (58.33%) male, Figure 1). Cognitive testing was offered to 208 (78.79%) at one to three months post-index stroke and 158 (59.85%) patients completed the assessment. Patients were followed-up for cognitive assessment at one (151, 57.20%) and three years’ post-index stroke (153, 57.95%). Those with cognitive follow-up had less severe stroke (one to three months), were younger, had more years of education (both one year) and differed on smoking status (three years) compared to those without cognitive follow-up. Additionally, those with cognitive follow-up at one year also had higher NART IQ and ACE-R scores at one to three months and those with cognitive follow-up at three years had higher one year ACE-R scores. There were no other significant differences in demographic or stroke characteristics at each time-point (Table 1). Across all time-points, NART IQ scores ranged from 92.18 to 127.70/127.70 and ACE-R scores ranged from 54 to 100/100 (Table 2).

### 3.2. Aim 1: Changes in Estimated Premorbid IQ and Current Cognitive Ability over Time

#### 3.2.1. NART IQ

Estimated premorbid IQ (NART IQ) changed significantly between all three time-points (Table 2, Figure 2a). Between one to three months and one year (*n* = 124), NART IQ significantly increased by a MD of 1.332 points (95% CI 0.536 to 2.129, *p* = 0.001). During this period, 77/124 (62%) improved, 35/124 (28%) declined, and 12/124 (10%) did not change.

Between one and three years (*n* = 101), NART IQ decreased significantly by −4.269 points (95% CI −5.124 to −3.414, *p* < 0.001). During this period, 12/101 (12%) improved, 82/101 (81%) declined, and 7/101 (7%) did not change.

Overall, between one to three months and three years (*n* = 100), NART IQ decreased significantly (−3.155, 95% CI −3.991 to −2.319, *p* < 0.001). The majority (78/100, 78%) of participants showed a decline.

The associations between time and NART IQ scores after controlling for baseline and stroke characteristics remained significantly positive (*b* = 1.453, 95% CI = 0.657 to 2.270, *p* < 0.001) at one to three months and one year, and significantly negative at one and three years (*b* = −4.474, 95% CI= −5.345 to −3.631, *p* < 0.001, Table 3)

#### 3.2.2. ACE-R

Current cognitive ability (ACE-R) remained generally more stable across all three time-points (Table 2 and Figure 2b). Between one to three months and one year (*n* = 135), ACE-R scores did not change significantly (0.289, 95% CI −0.488 to 1.065, *p* = 0.463). During this period, 61/135 (45%) patients improved, 54/135 (40%) declined, and 20/135 (15%) did not change.

Between one and three years (*n* = 106), ACE-R scores declined significantly (−1.047, 95% CI −2.082 to −0.012, *p* = 0.047). During this period, 42/106 (40%) patients improved, 57/106 (54%) declined, and 7/106 (7%) did not change.

Overall, between one to three months and three years (*n* = 101), 48/101 (48%) improved, 42/101 (16%) declined, and 11/101 (10%) did not change, however the overall change was not significant (0.178, 95% CI −1.354 to 0.997, *p* = 0.764).

The associations between time and ACE-R scores remained unchanged after controlling for baseline and stroke characteristics: Positive, but non-significant, at one to three months and one year (*b* = 0.403, 95% CI = −0.375 to 1.195, *p* = 0.312) and significantly negative at one and three years (*b* = −1.028, 95% CI = −2.039 to −0.003, *p* = 0.047, Table 3).

#### 3.2.3. Sensitivity Analysis

Including only patients with cognitive data at all three time-points did not change the patterns of NART IQ (*n* = 90) and ACE-R (*n* = 96) scores (Appendix A and Figure 2c,d).

#### 3.2.4. The Relationship between Age and Estimated Premorbid IQ (NART IQ) and Current Cognitive Ability (ACE-R)

The relationship between age with NART IQ and ACE-R scores were in different directions. Age was positively associated with NART IQ at one to three months post-stroke (i.e., there was a trend for those with older age at entry into study having higher NART IQ), however this association was non-significant (*b* = 0.069, 95% CI = −0.048 to 0.181, *p* = 0.24, Figure 3). For ACE-R, the relationship between age and ACE-R was negative (i.e., older age at entry into study was associated with lower current cognitive ability). This association was significant (*b* = −0.210, 95% CI = −0.314 to −0.107, *p* < 0.001, Figure 3).

### 3.3. Aim 2: Predictors of Changes between Time-Points

#### 3.3.1. NART IQ

More years of education was associated with increases in NART IQ between one to three months and one year (*b* = 1.165, 95% CI = 0.721 to 1.608), *p* < 0.001), and smaller decreases between one and three years’ (*b* = 1.046, 95% CI = 0.598 to 1.495, *p* < 0.001). There were no other significant interactions suggesting that no other baseline demographics or stroke characteristics were associated with changes in NART IQ (Table 4 and Appendix A).

#### 3.3.2. ACE-R

Older age was associated with declines in ACE-R scores between one to three months (*b* = −0.202, 95% CI = −0.311 to −0.093, *p* < 0.001), and one and three years’ (*b* = −0.137, 95% CI = −0.255 to −0.019, *p* = 0.024). Increases in ACE-R were associated with higher NART IQ (one to three months and one year: *b* = 0.345, 95% CI = 0.229 to 0.461, *p* < 0.001, one and three years: *b* = 0.416, 95% CI = 0.286 to 0.546, *p* < 0.001) and more education (one to three months and one year: *b* = 0.603, 95% CI = 0.229 to 0.977, *p* = 0.002, and one and three years: *b* = 0.501, 95% CI = 0.113 to 0.888, *p* = 0.012). In addition, increases in ACE-R between one to three months and one year were associated with having hypertension (*b* = 2.943, 95% CI = 0.605 to 5.280, *p* = 0.014). There were no other significant interactions (Table 5 and Appendix A).

## 4. Discussion

We assessed patients with minor ischaemic stroke over a follow-up period of three years assessing the stability of both estimated premorbid IQ and current cognitive ability using established measures. To our knowledge, this is the first study to assess the stability of the NART post-stroke and one of a few studies to assess current cognition to three years post-stroke. Our findings suggest that post-stroke cognitive change is complex. We show fluctuations in NART IQ scores with a significant increase between one to three months and one year, followed by a significant decrease between one and three years’ post-index stroke. For ACE-R scores, we found that scores remained mostly stable between first assessment at between one to three months and second assessment at one year, followed by a small, but significant, decrease by the third assessment at three years. Adjusting for baseline demographic and stroke characteristics did not change these trajectories. Increases in both NART IQ and ACE-R scores post-stroke were associated with more education. Additionally, higher NART IQ and older age were associated with changes in ACE-R scores (increases and decreases respectively). Despite including relevant outcome predictors (e.g., age, stroke severity, WMH volume etc.), the poor model fit may reflect the lack of relevant variables that contribute to cognitive ability scores. This highlights the need for future research closely examining variables associated with changes in both estimated premorbid and current cognitive ability to better understand post-stroke cognitive trajectories.

The stability of the NART in other conditions has shown mixed findings. Although, the NART has been shown to be a stable estimate of premorbid IQ in healthy individuals [19] and patients with schizophrenia (mean age 35 years) [11] over one and 7.5 years respectively, small but significant decreases have been shown in healthy older individuals [20] and those with dementia [21]. Furthermore, improvements in NART performance has also been shown in traumatic brain injury (TBI) [22,23] demonstrating the complex interaction of disease variables, age, education and time post-injury affecting NART performance.

Symptoms such as post-stroke fatigue and speech disturbances are common [24,25] and may affect NART performance shortly after stroke. Symptom improvement may explain the initial increase in NART IQ, however these variables would also affect ACE-R performance, which is not seen here. Furthermore, factors associated with declining NART performance, such as increasing rates of dementia, were not frequently observed in our study and recurrent stroke between the one and three year follow-up was uncommon (*n* = 12, 8%).

We show only a small, albeit significant, decrease in current cognitive ability (ACE-R) between one and three years’. Although cognitive ability declines with increasing age [26,27] and post-stroke dementia is common [28], studies show mixed post-stroke cognitive trajectories [29]. Improvements in global cognition are seen in some studies with short follow-up periods (up to three months) [30], while those with longer follow-up periods (up to six years) report declines [31].

We show a significant negative relationship between age and current cognitive ability, however the association between NART IQ and age is less clear. Research in individuals from the general population has shown better crystallized intelligence in older adults [32], although some studies have demonstrated a negative relationship between age and NART [13]. We show a weak, non-significant, positive association such that individuals entered into the study having had a stroke at older ages had marginally higher NART IQ, although this association was non-significant. If true in other studies of post-stroke cognition, then age is a source of confounding and variation in premorbid IQ may account for poor prediction of post-stroke cognitive impairment.

We did not find that adjusting for demographic and stroke characteristics, including age, altered the association between time and changes in ACE-R and NART IQ. This suggests that other unidentified factors are contributing to this pattern of results. Future research should focus on the identification of factors influencing longitudinal changes in premorbid IQ in stroke patients to enable the accurate prediction and identification of those at risk of poor post-stroke cognitive outcomes.

### Strengths and Limitations

Although the MSSII achieved good follow-up at all three time-points, we found some evidence of possible participation bias as those with follow-up had better previous ACE-R and NART IQ scores. This may have contributed to the overall stability of current cognitive ability, however the trends in both NART IQ and ACE-R scores remained the same when only those with cognitive data at all time-points were examined and when accounting for baseline demographic and stroke characteristics. Furthermore, we used linear mixed effect models to examine factors associated with change which is unaffected by missing data points and accounts for different starting points (i.e., someone who decreases from NART IQ of 120 to 117 is not necessarily the same as someone who decreases from 90 to 97).

Cognitive assessment were administered by one rater at one to three months and one year and different raters at three years’ which may have resulted in some variability. This is particularly true with the NART, which relies on the subjective assessment of correct pronunciation, however the inter-rater reliability of the NART has previously shown to be good [22,33,34] and all researchers administering the assessments had undergone the same training.

## 5. Conclusions

Overall, we show an interesting dichotomy with declines in estimates of premorbid IQ, while current cognitive ability scores remain more stable. These findings suggest caution in the use of the NART to estimate premorbid IQ after stroke, although its use is more beneficial than having no measure. Future research in post-stroke cognition should try to account for premorbid IQ since it is a major predictor of post-stroke cognition and health in general and should closely examine what factors influence changes in premorbid IQ following stroke so that more reliable estimates can be obtained.

## Figures and Tables

**Figure 1 brainsci-09-00117-f001:**
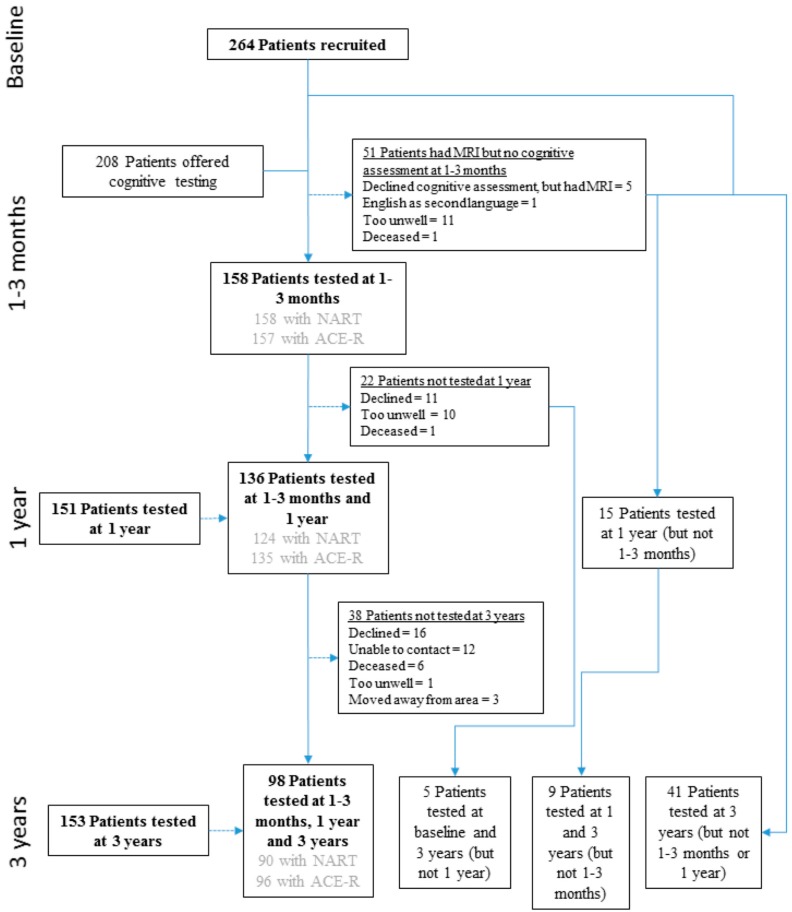
Flow chart showing number of participants with cognitive assessment (estimated premorbid, current, or both) at each time-point in the Mild Stroke Study II (MSSII).

**Figure 2 brainsci-09-00117-f002:**
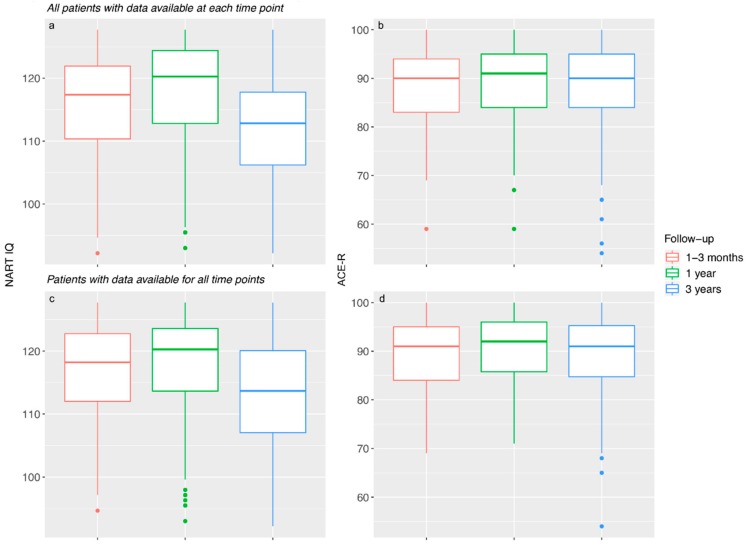
NART IQ and ACE-R scores by time-point (**a**) NART IQ scores by time-point for patients with data available at each time-point. (**b**) ACE-R scores by time-point for patients with data available at each time-point. (**c**) NART-IQ scores by time-point for patients with data available at all three time-points. (**d**) ACE-R scores by time-point for patients with data available at all three time-points.

**Figure 3 brainsci-09-00117-f003:**
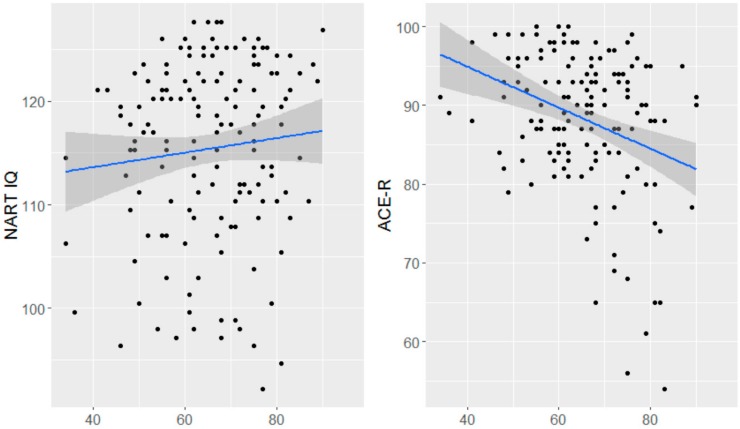
The relationship between NART IQ and ACE-R with age at first assessment at between one to three months.

**Table 1 brainsci-09-00117-t001:** Baseline characteristics of patients with cognitive data by time-point.

	One to Three Months (*n* = 158)		1 Year (*n* = 151)		3 Years (*n* = 153)	
	With vs. Without Cognitive Data		With vs. Without Cognitive Data		With vs. Without Cognitive Data	
**Demographics**						
Age at index stroke	65.09 ± 11.82	W = 9544.5	65.01 ± 11.02	W = 9845 *	65.50 ± 11.18	W = 9476.5
Sex: Male	93 (58.86%)	*χ*^2^ = 0.007	93 (61.59%)	*χ*^2^ = 1.242	83 (54.25%)	*χ*^2^ = 2.115
Years of Education ^a^	11.71 ± 2.94	W = 876.5	11.88 ± 3.02	W = 1098.5 *	12.04 ± 3.11	W = 2633
Smoker:		*χ*^2^ = 2.954		*χ*^2^ = 7.551		*χ*^2^ = 12.256 *
Yes	48 (30.38%)		42 (27.81%)		43 (28.29%)	
No	56 (35.44%)		58 (38.41%)		61 (40.13%)	
Ex-smoker (for more than 1 year)	47 (29.75%)		44 (29.14%)		44 (28.95)	
Ex-smoker (for less than 1 year)	7 (4.43%)		7 (4.64%)		4 (2.63)	
**Stroke Characteristics**						
Stroke subtype:		*χ*^2^ = 0.287		*χ*^2^ = 0.998		*χ*^2^ = 0.224
Cortical	90 (56.96%)		88 (58.28%)		87 (56.86%)	
Lacunar	68 (43.04%)		63 (41.72%)		66 (43.14%)	
Stroke severity (NIHSS)	1.00 (1.00)	W = 9853.5 *	1.00 (1.00)	W = 9319.5	1.00 (2.00)	W = 7709
**Cognitive Variables**						
ACE-R	88.09 ± 8.17	-	88.93 ± 7.98	W = 815 **^,b^	88.32 ± 8.92	W = 2512/W = 1864 *^,c^
NART IQ	115.40 ± 8.60	-	117.56 ± 8.61	W = 1031 *^,b^	112.28 ± 8.40	W = 2549.5/W = 1998.5 ^c^

^a^ Collected at the 1-year follow-up; ^b^ baseline scores; ^c^ baseline/1-year follow-up scores; * *p* < 0.05; ** *p* < 0.001; W = Wilcoxon rank-sum test; *χ*^2^ = chi-square test; ACE-R = Addenbrooke’s Cognitive Examination-Revised; NART IQ = Premorbid IQ estimated using The National Adult Reading Test.

**Table 2 brainsci-09-00117-t002:** NART IQ and ACE-R scores at and between each time-point for all patients with data available at each time-point.

	Scores				Change in Scores		
	1–3 Months	1 Year	3 Years		1–3 Months to 1 Year	1 to 3 Years	1–3 Months to 3 Years
	**NART IQ**						
*n*	154	140	153	*n*	124	101	100
Mean ± SD (range)	115.40 ± 8.60 (92.18–127.70)	117.56 ± 8.61 (93.08–127.70)	112.28 ± 8.40 (92.18–127.70)	Mean change (95% CI)	1.322 (0.54 to 2.13) **	−4.269 (−5.12 to −3.41) **	−3.155 (−3.99 to 2.32) **
				Decreased *n* (%)	35 (28.23%)	82 (81.19%)	78 (78.00%)
				Increased *n* (%)	77 (62.10%)	12 (11.88%)	15 (15.00%)
				No change *n* (%)	12 (9.68%)	7 (6.93%)	7 (7.00%)
	**ACE-R**						
*n*	157	151	151	*n*	135	106	101
Mean ± SD (range)	88.09 ± 8.17 (59.00–100.0)	88.93 ± 7.98 (59.00–100.0)	88.32 ± 8.92 (54.00–100.0)	Mean change (95% CI)	0.289 (−0.49 to 1.07)	−1.047 (−2.08 to −0.01) *	−0.178 (−1.35 to 1.00)
No impairment *n* (%) ^a^	91 (57.96%)	91 (60.26%)	85 (56.29%)	Decreased *n* (%)	54 (40.00%)	57 (53.77%)	42 (41.58%)
Mild impairment *n* (%) ^b^	28 (17.83%	28 (18.54%)	35 (23.18%)	Increased *n* (%)	61 (45.19%)	42 (39.62%)	48 (47.52%)
Severe impairment *n* (%) ^c^	38 (24.20%)	32 (21.19%)	31 (20.53%)	No change *n* (%)	20 (14.81%)	7 (6.60%)	11 (10.89%)

* *p* < 0.05; ** *p* < 0.001; ^a^ ACE-R scores ≥ 89; ^b^ ACE-R scores 83–88; ^c^ ACE-R scores ≤ 82.

**Table 3 brainsci-09-00117-t003:** Linear mixed effects model showing the associations between time and NART IQ and ACE-R adjusted for baseline characteristics.

	1–3 Months and 1 Year			1–3 Years		
	*B*	95% CI	*p*	*B*	95% CI	*p*
	**NART IQ**					
Time (years)	*1.453*	*(0.657, 2.270)*	*<0.001*	*−4.474*	*(−5.345, −3.631)*	*<0.001*
Age	0.100	(−0.026, 0.225)	0.118	0.099	(−0.039, 0.237)	0.170
Sex (male)	1.232	(−1.222, 3.587)	0.316	1.497	(−1.005, 3.999)	0.253
Baseline Fazekas	0.166	(−0.639, 0.973)	0.692	−0.002	(−0.870, 0.865)	0.996
Stroke subtype (lacunar)	1.047	(−1.401, 3.495)	0.413	0.893	(−1.721, 3.508)	0.514
NIHSS	−0.214	(−1.306, 0.877)	0.707	0.032	(−1.120, 1.182)	0.958
Hypertension (yes)	2.815	(0.222, 5.409)	0.075	2.304	(−0.466, 5.075)	0.112
Smoker (yes)	−0.536	(−3.158, 2.086)	0.696	−1.574	(−4.397, 1.251)	0.287
Years of education	1.287	(0.877, 1.697)	<0.001	1.193	(0.775, 1.612)	<0.001
	**ACE-R**					
Time (years)	*0.403*	*(−0.375, 1.195)*	*0.312*	*−1.028*	*(−2.039, −0.003)*	*0.047*
Age	−0.194	(−0.308, −0.080)	0.001	−0.162	(−0.292, −0.032)	0.017
Sex (male)	0.362	(−1.758, 2.479)	0.743	0.392	(−1.947, 2.732)	0.749
Baseline Fazekas	−0.316	(−1.039, 0.410)	0.403	−0.692	(−1.499, 0.114)	0.101
Stroke subtype (lacunar)	0.728	(−1.474, 2.927)	0.526	1.376	(−1.070, 3.825)	0.282
NIHSS	−0.421	(−1.398, 0.556)	0.410	−0.316	(−1.382, 0.749)	0.571
Hypertension (yes)	3.709	(1.380, 6.037)	0.002	3.465	(0.864, 6.066)	0.011
Smoker (yes)	−0.355	(−2.716, 2.009)	0.772	−0.182	(−2.819, 2.461)	0.895
Years of education	0.952	(0.581, 1.322)	<0.001	1.009	(0.615, 1.404)	<0.001

NART IQ = Premorbid IQ estimated using the National Adult Reading Test; ACE-R = Addenbrooke’s Cognitive Examination-Revised. Results in italics highlight the estimates of interest showing that the relationships between time and changes in NART-IQ and ACE-R remain after adjusting for baseline characteristics.

**Table 4 brainsci-09-00117-t004:** Linear mixed effects models showing factors associated with changes in NART IQ at each time-point.

Variable	Variation in NART IQ with Variable at:			Change in NART IQ between:			Change in NART IQ with Variable at:		
	1–3 Months			1–3 Months and 1 Year			1 Year		
	*B*	95% CI	*p*	*B*	95% CI	*p*	*B*	95% CI	*p*
Age	0.095	(−0.038, 0.229)	0.160	1.339	(−3.545, 6.223)	0.588	0.097	(−0.039, 0.233)	0.160
Sex	1.546	(−0.998, 4.090)	0.231	1.878	(0.582, 3.173)	0.005	0.847	(−1.760, 3.453)	0.521
Baseline Fazekas	0.402	(−0.458, 1.262)	0.357	2.954	(1.306, 4.602)	<0.001	−0.123	(−1.004, 0.758)	0.782
Stroke subtype (lacunar)	0.704	(−1.926, 3.334)	0.597	1.122	(0.035, 2.210)	0.043	1.454	(−1.238, 4.145)	0.287
NIHSS	−0.340	(−1.513, 0.833)	0.567	1.163	(0.066, 2.260)	0.038	−0.062	(−1.253, 1.130)	0.918
Hypertension (yes)	2.970	(0.162, 5.778)	0.038	1.695	(0.121, 3.269)	0.035	2.642	(−0.222, 5.506)	0.070
Smoker (yes)	−0.256	(−3.073, 2.562)	0.858	1.672	(0.692, 2.651)	0.001	−0.957	(−3.836, 1.923)	0.512
Years of education	1.400	(0.958, 1.842)	<0.001	4.277	(1.060, 7.494)	0.010	1.165	(0.721, 1.608)	<0.001
	**1 Year**			**1 and 3 Years**			**3 Years**		
Age	0.098	(−0.061, 0.257)	0.223	−2.107	(−4.706, 0.492)	0.111	0.0963	(−0.050, 0.243)	0.194
Sex	1.214	(−1.866, 4.293)	0.436	−2.327	(−2.997, −1.657)	<0.001	1.3677	(−1.345, 4.081)	0.319
Baseline Fazekas	−0.242	(−1.281, 0.797)	0.644	−2.633	(−3.551, −1.716)	<0.001	−0.103	(−1.035, 0.828)	0.826
Stroke subtype (lacunar)	2.980	(−0.160, 6.119)	0.063	−1.723	(−2.283, −1.164)	<0.001	1.842	(−0.979, 4.663)	0.198
NIHSS	0.213	(−1.166, 1.592)	0.760	−2.129	(−2.727, −1.531)	<0.001	0.116	(−1.122, 1.354)	0.852
Hypertension (yes)	2.012	(−1.383, 5.406)	0.242	−2.36	(−3.215, −1.505)	<0.001	2.176	(−0.815, 5.168)	0.152
Smoker (yes)	−1.051	(−4.446, 2.344)	0.540	−2.152	(−2.658, −1.646)	<0.001	−1.369	(−4.393, 1.656)	0.371
Years of education	0.873	(0.375, 1.372)	0.001	−4.332	(−6.003, −2.661)	<0.001	1.046	(0.598, 1.495)	<0.001

All estimates are adjusted for the other variables in the model. Each row represents an individual, but strongly related, overlapping model. Full tables for each model can be found in Appendix A.

**Table 5 brainsci-09-00117-t005:** Linear mixed effects models showing factors associated with changes in current ACE-R scores at each time point.

Variable	Variation in ACE-R with variable at:			Change in ACE-R between:			Change in ACE-R with variable at:		
	1–3 Months			1–3 Months and 1 Year			1 Year		
	*B*	95% CI	*p*	*B*	95% CI	*p*	*B*	95% CI	*p*
Age	−0.239	(−0.345, −0.133)	<0.001	−2.191	(−7.337, 2.954)	0.401	−0.202	(−0.311, −0.093)	<0.001
Sex (male)	−0.370	(−2.417, 1.677)	0.721	−0.001	(−1.389, 1.387)	0.999	−0.030	(−2.150, 2.090)	0.977
Baseline Fazekas	−0.221	(−0.915, 0.473)	0.529	0.207	(1.597, 2.011)	0.820	−0.222	(−0.937, 0.494)	0.541
Stroke subtype (lacunar)	−0.327	(−2.433, 1778)	0.759	−0.020	(−1.177, 1.137)	0.973	0.191	(−1.995, 2.377)	0.863
NIHSS	−0.609	(−1.541, 0.323)	0.198	−0.536	(−1.683, 0.612)	0.357	0.115	(−0.840, 1.070)	0.812
Hypertension (yes)	2.761	(0.499, 5.033)	0.018	0.073	(−1.601, 1.746)	0.932	2.943	(0.605, 5.280)	0.014
Smoker (yes)	0.388	(−1.859, 2.635)	0.733	0.595	(−0.452, 1.642)	0.263	−0.835	(−3.165, 1.494)	0.479
NART IQ	0.290	(0.171, 0.408)	0.000	−6.217	(−18.303, 5.869)	0.310	0.345	(0.229, 0.461)	<0.001
Years of education	0.407	(0.026, 0.788)	0.037	−2.157	(−5.616, 1.302)	0.219	0.603	(0.229, 0.977)	0.002
	**1 Year**			**1 and 3 Years**			**3 Years**		
Age	−0.054	(−0.196, 0.087)	0.447	5.771	(2.560, 8.981)	0.001	−0.137	(−0.255, −0.019)	0.024
Sex (male)	1.016	(−1,928, 3.960)	0.495	0.803	(−0.114, 1.720)	0.085	0.370	(−1.882, 2.622)	0.745
Baseline Fazekas	−0.123	(−1.088, 0.842)	0.801	1.250	(0.018, 2.482)	0.047	−0.407	(−1.167, 0.353)	0.291
Stroke subtype (lacunar)	−1.008	(−3.990, 1.974)	0.504	<0.001	(−0.783, 0.783)	0.999	0.022	(−2.291, 2.335)	0.985
NIHSS	0.227	(−1.063, 1.517)	0.727	0.683	(−0.138, 1.504)	0.102	−0.005	(−1.011, 1.000)	0.992
Hypertension (yes)	2.378	(−0.875, 5.632)	0.150	0.444	(−0.709, 1.598)	0.446	2.367	(−0.108, 4.843)	0.061
Smoker (yes)	−1.491	(−4.670, 1.689)	0.354	0.089	(−0.619, 0.796)	0.804	−0.226	(−2.697, 2.245)	0.856
NART IQ	0.380	(0.211, 0.549)	<0.001	−3.704	(−11.427, 4.020)	0.343	0.416	(0.286, 0.546)	<0.001
Years of education	0.548	(0.064, 1.032)	0.027	1.019	(−1.312, 3.350)	0.387	0.501	(0.113, 0.888)	0.012

All estimates are adjusted for the other variables in the model. Each row represents an individual, but strongly related, overlapping model. Full tables for each model can be found in Appendix A.

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
