# Peer review of "Stability of Estimated Premorbid Cognitive Ability over Time after Minor Stroke and Its Relationship with Post-Stroke Cognitive Ability"

_brainsci, 2019, doi:10.3390/brainsci9050117_

Round 1
Reviewer 1 Report
This study has 2 purposes:
1) Assess the validity of the NART for the premorbid IQ after stroke. Authors take into account the relationship between age, estimated premorbid IQ and cognitive ability from 1 to 3 months post-stroke.
2) Make a link between changes in NART IQ and current cognition scores with baseline demographic or stroke characteristics.
The manuscript is well written. This study is relevant for the stroke patient care. Nevertheless, few points need to be improved:
Introduction:
- Authors need to define more precisely the peak adult intelligence, and why is it also called “premorbid IQ”. It is not sufficiently explained in the introduction.
- The difference between the 2 aims also needed to be clearer.
Materials and Method:
- Why patients with cognitive impairments were excluded? What the limits of the cognitive deficit severity? The rationale is not clear when reading the manuscript.
- The NART and the ACE-R are not described. I believe that it might be important to indicate important information about the procedure of the tests. What does that mean “….completed by a trained researcher”? Trained for what? It also should be described to better understand what was done. The procedure might influence results.
Results and Discussion:
No special comments for these parts. Nevertheless, in result section, where are the point 3.2? I think it is a mistake.
Author Response
We thank the reviewer for their comments. Please see comments below (in red) for responses to specific comments.
Comment 1 (Authors need to define more precisely the peak adult intelligence, and why is it also called “premorbid IQ”. It is not sufficiently explained in the introduction):
We use 'peak adult
intelligence' since the term 'premorbid' is sometimes taken to mean 'cognitive
ability immediately prior to the stroke', but since cognition may have been
declining prior to the stroke (indeed cognitive decline is a risk factor for
stroke in older people), cognitive status immediately prior to the stroke
cannot be used as a measure of peak adult ability. Hence the use of NART. We have now updated this section of the introduction to elaborate and provide a reference for the definition of peak adult intelligence/premorbid IQ. This section now reads as follows:
“Cognition in childhood determines cognitive ability in later life in healthy persons[1]and influences cognition in those with dementia.[2] Therefore it is important to consider peak adult intelligence (IQ), also referred to as ‘premorbid IQ’, when assessing cognitive ability in later life to allow for the examination of cognitive decline.[3] Premorbid IQ refers to an individual’s level of intellectual functioning prior to declines associated with ageing and/or neurological events.[4]”
Comment 2 (The difference between the 2 aims also needed to be clearer):
We have amended the aims to emphasise that aim 1 focuses on the coefficients for time and whether this changes when we control for demographic and stroke characteristics and aim 2 focuses on predictors of change in NART IQ and ACER scores. The aims now read:
“We aimed to investigate:
1. Whether the NART is a valid test of premorbid IQ after stroke by examining the relationship between time after stroke with changes in NART IQ, and current cognitive ability (i.e. how do estimated premorbid IQ and current cognitive ability scores change between follow-up time-points). We will also examine whether the relationships with time remain after controlling for baseline demographic and stroke characteristics. Due to the obvious relationship between age and time, we examine the relationship between age, estimated premorbid IQ and cognitive ability at one to three months post-stroke.
2. Whether certain baseline demographic or stroke characteristics predict changes in NART IQ and current cognition scores between two time-points post-stroke.”
Comment 3 (Why patients with cognitive impairments were excluded? What the limits of the cognitive deficit severity? The rationale is not clear when reading the manuscript.):
At baseline, participants who lacked capacity were excluded. Capacity to consent was assessed clinically. Lacking capacity is an indicator of quite definite cognitive impairment as you can still have capacity and have mild cognitive impairment. As the exclusion was based on capacity rather than specific cognitive ability, the study captured a range of abilities after stroke as long as they were cognitive able to give consent. The sentence has been updated to read:
“Patients who were unable to consent (e.g. lacked capacity due to cognitive impairment), medically unlikely to participate in long-term follow up and had aphasia were excluded (details previously described) [7,15].”
Comment 4 (The NART and the ACE-R are not described. I believe
that it might be important to indicate important information about the
procedure of the tests. What does that mean “….completed by a trained
researcher”? Trained for what? It also should be described to better understand
what was done. The procedure might influence results.)
The NART has been described in the introduction however we have edited this and included a couple of additional sentences for the NART and ACE-R in the methods section (see below).
“The NART consists of 50 phonetically irregular words which participants read aloud. The ACE-R is a multi-domain cognitive screening tool used to identify those with cognitive impairment."
All screening tools used in this study can be administered without formal training however all researchers involved were trained in the assessment of patients and had experience in the administration of the cognitive tools used in this study. This sentence now reads:
“Assessments were completed by trained researchers, experienced in the administration of these neuropsychological screening tools."
Comment 5 (No special comments for these parts. Nevertheless, in
result section, where are the point 3.2? I think it is a mistake.
Thank you for pointing this out. We have amended the heading numbers in this section
Reviewer 2 Report
This is an interesting paper showed that fluctuations in estimated premorbid IQ and minor changes in current cognitive ability after stroke. I have two minor questions
(1) Any significant difference between male and female?
(2) The data could be further analyzed to uncover more information. For example, the patients could be further divided into several groups by age and compare the stroke-induced cognitive impairment at different stage.
Author Response
We thank the reviewer for their comments. Please see comments below (in red) for responses to specific comments.
Comment 1 (Any significant difference between male and female?):
We have performed additional analyses to include sex and found that including gender did not change the relationship between NART IQ and ACE-R with time (aim 1). In aim 2, sex was not associated with changes in both ACE-R and NART IQ scores between time points. All relevant tables have now been updated to include sex.
Comment 2 (The data could be further analyzed to uncover more information. For example, the patients could be further divided into several groups by age and compare the stroke-induced cognitive impairment at different stage.)
We chose to include age as a continuous variable in our model as it is more sensitive than using a categorical variable. Although clinically it may be interesting to examine categories of age, it is not within the scope of this paper.